# Neural Probabilistic Logic Learning for Knowledge Graph Reasoning

## Abstract

Knowledge graph (KG) reasoning is a task that aims to predict unknown facts based on known factual samples. Reasoning methods can be divided into two categories: rule-based methods and KG-embedding based methods. The former possesses precise reasoning capabilities but finds it challenging to reason efficiently over large-scale knowledge graphs. While gaining the ability to reason over large-scale knowledge graphs, the latter sacrifices reasoning accuracy. This paper aims to design a reasoning framework called Neural Probabilistic Logic Learning(NPLL) that achieves accurate reasoning on knowledge graphs. Our approach introduces a scoring module that effectively enhances the expressive power of embedding networks. We strike a balance between model simplicity and reasoning capabilities by incorporating a Markov Logic Network based on variational inference. We empirically evaluate our approach on several benchmark datasets, and the experimental results validate that our method substantially enhances the accuracy and quality of the reasoning results. paragraph.

## 1 Introduction

Knowledge representation has long been a fundamental challenge in artificial intelligence. Knowledge graphs, a form of structured knowledge representation, have gained significant traction in recent years due to their ability to capture rich semantics and facilitate reasoning over large-scale data. Compared to conventional approaches such as semantic networks and production rules, knowledge graphs offer a more expressive and scalable representation of entities and their relationships in a graph-based formalism. This structured representation not only assists human comprehension and reasoning but also enables seamless integration with machine learning techniques, facilitating a wide range of downstream applications.

One prominent line of research in knowledge graph reasoning revolves around embedding-based methods. These techniques aim to map the elements of a knowledge graph into a low-dimensional vector space, capturing the underlying associations between entities and relations through numerical representations. While this approach has demonstrated promising results, it suffers from inherent limitations, including low interpretability, suboptimal performance on long-tail relations, and challenges in capturing complex semantic information and logical relationships.

Alternatively, rule-based knowledge reasoning methods operate by extracting logical rules from the knowledge graph, typically in the form of first-order predicate logic, and performing inference based on these rules. However, these methods often face challenges stemming from the vast search space and limited coverage of the extracted rules. Markov Logic Networks (MLNs) (Richardson & Domingos, 2006) have been proposed as a principled framework for combining probabilistic graphical models with first-order logic, enabling the effective integration of rules and embedding methods for more accurate reasoning.

In this paper, we seek to develop a method that can better leverage the outputs of embedding networks to support knowledge graph reasoning. To this end, we propose a novel reasoning framework called Neural Probabilistic Logic Learning (NPLL). NPLL introduces a scoring module that efficiently utilizes knowledge graph embedding data, enhancing the training process of the entire framework. Our method, illustrated in Figure 1, makes the following key contributions:

**Large-scale KG reasoning capability**: NPLL effectively handles reasoning tasks in large-scale KGs. Experimental results demonstrate its performance in knowledge bases containing millions of facts.

**Efficient Reasoning and Learning**: NPLL can be viewed as an inference network for MLNs, extending MLN inference to larger-scale knowledge graph problems.

**Tight Integration of Logical Rules and Data Supervision**: NPLL can leverage prior knowledge from logical rules and supervision from structured graph data.

**Balance between Model Size and Reasoning Capability**: Despite its compact architecture and relatively fewer parameters, NPLL demonstrates remarkable reasoning capabilities, sufficient to capture the intricate relationships and semantics within knowledge graphs. Even in data-scarce scenarios where the available dataset size is relatively small, NPLL can achieve a high level of reasoning performance, making it well-suited for practical applications with limited labeled data.

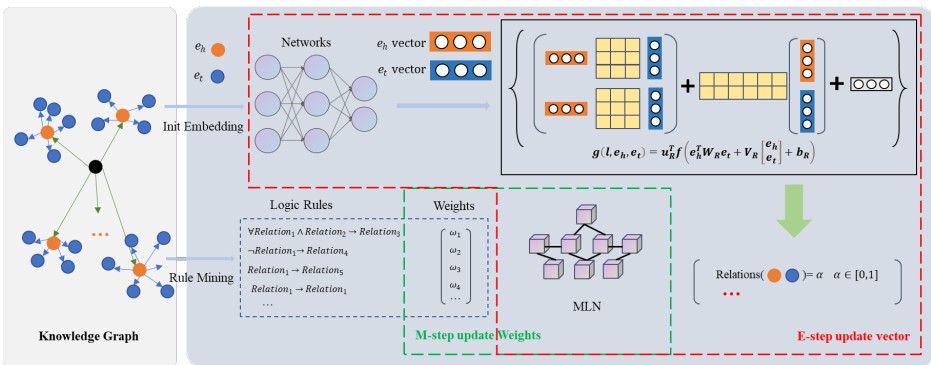

Figure 1: Visualization of Neural Probabilistic Logic Learning (NPLL)

## 2 RELATED WORK

One prominent category of methods for knowledge graph reasoning is rule-based approaches. These methods leverage logical rules, generally defined as B→A, where A is the target fact, and B can be considered a set of condition facts. Facts are composed of predicates and variables. To better utilize these symbolic features, methods like AMIE(Galárraga et al., 2013), RuleN(Meilicke et al., 2018), and RLvLR(Omran et al., 2019) employ rule mining tools to extract logical rules from knowledge graphs for reasoning. Approaches like KALE(Guo et al., 2016), RUGE(Guo et al., 2018), and IterE(Zhang et al., 2019b) started combining logical rules with embedding learning to construct joint knowledge graph reasoning models. Additionally, NeuralLP(Yang et al., 2017) proposed an end-to-end differentiable method to effectively learn the parameters and structures of logical rules in knowledge graphs. NeuralLP-num-lp(Wang et al., 2019) combined summation operations and dynamic programming with NeuralLP, which can be used to learn numerical regulations better. Simultaneously, DRUM(Sadeghian et al., 2019) introduced a rule-based end-to-end differentiable model. Then, pLogicNet designed a probabilistic logic neural network (Qu & Tang, 2019), demonstrating exemplary reasoning performance. Building on this, ExpressGNN(Zhang et al., 2020b) achieved more efficient reasoning by fine-tuning the GNN model. DiffLgic(Shengyuan et al., 2024) designed a differential framework to improve reasoning efficiency and accuracy for large knowledge graphs. NCRL(Cheng et al., 2023) proposed an end-to-end neural method that recursively leverages the compositionality of logical rules to enhance systematic generalization. In contrast to these approaches, our proposed NPLL framework is significantly more effective for knowledge graph reasoning.

Another category of approaches for knowledge graph reasoning is embedding-based methods. These techniques primarily represent entities and relations using vector embeddings. Knowledge graph reasoning is achieved by defining various scoring functions to model different reasoning processes. For instance, methods like TransE(Bordes et al., 2013), TransH(Wang et al., 2014), TransR(Lin et al., 2015), TransD(Ji et al., 2016), TranSparse(Ji et al., 2015), TransRHS(Zhang et al., 2020a), RotatE(Sun et al., 2019) project entities and relations into vector spaces, transforming computations between facts into vector operations. The essential scoring function is the difference between the

head entity-relation vector and the tail entity vector. Rescal(Nickel et al., 2011), DistMult(Yang et al., 2014), ComplEx(Trouillon et al., 2016), HolE(Nickel et al., 2016), analog(Liu et al., 2017), SimplE(Kazemi & Poole, 2018), QuatE(Zhang et al., 2019a), DualE(Cao et al., 2021), HopfE(Bastos et al., 2021), LowFER(Amin et al., 2020), QuatRE(Nguyen et al., 2022) represent each fact in the knowledge graph as a three-dimensional tensor, decomposed into a combination of low-dimensional entity and relation vectors. They use vector matrices to represent the latent semantics of each entity and relation. The primary scoring function is the product of the head entity, relation, and tail entity. Methods like SME(Bordes et al., 2014), NTN(Socher et al., 2013), and NAM(Liu et al., 2016) employ neural networks to encode entities and relations into high-dimensional spaces. ConvE(Dettmers et al., 2018) first introduced 2D convolutional layers for reasoning. RGCN(Schlichtkrull et al., 2018), NBF-net(Zhu et al., 2021), and RED-GNN(Zhang & Yao, 2022) use graph neural networks to aggregate neighbor node information and decoders as scoring functions. However, these embedding-based methods often sacrifice interpretability and prediction quality. In contrast, our proposed NPLL framework significantly improves the quality of reasoning results while more properly handling reasoning problems through the principled integration of logical rules.

## 3 PRELIMINARY

A knowledge graph is a graph-structured model composed of triplets, where the entities in the triplets are nodes and the relations are edges. Given a known knowledge graph $K = (E, L, F)$, where $E = \{e_1, e_2, \ldots, e_M\}$ represents a set of M entities, with entities typically referring to person names, objects, locations and proper nouns; $L = \{l_1, l_2, \ldots, l_N\}$ represents a set of N relations; $F = \{f_1, f_2, \ldots, f_S\}$ represents a set of known facts involving entities from E and relations from L, where fi can be described as $f_i = \{e_h, l, e_t\}, e_h, e_t \in E, l \in L$, indicating $e_h$ has a relation $l$ with $e_t$, or can be written as $l(e_h, e_t)$, where $l$ is treated as a predicate and $e_h$ and $e_t$ as constants.

We now introduce the predicate logic representation, where each relation in the relation set is represented as a function $l(x, y)$, with $x$ and $y$ having the domain $E$, and $l(x, y)$ being directed, so $l(x, y)$ and $l(y, x)$ are different. For example, $l(x, y) := S(Tom, basketball)$ ($S$ denotes proficient sport), indicates that Tom's proficient sport is basketball, which clearly cannot be expressed as $S(basketball, Tom)$.

Using the predicate logic representation, new facts can be inferred through logical deduction, e.g., $S(Tom, basketball) \wedge F(Tom, John) \Rightarrow S(John, basketball)$ ($F$ denotes being friends). If variables replace the constant entities in the above formula, it is called a rule, generally represented as: $Pred_1(x, y_1) \wedge Pred_2(y_1, y_2) \wedge \ldots Pred_n(y_{n-1}, z) \Rightarrow Pred(x, z)$ $n \geq 1$, where $x, y_i', z$ are all variables. $Pred(A, B)$ is called an atom, with $A$ and $B$ being the subject and object or the head and tail entities in the triplet. $Pred(x, z)$ is the head atom; the rest are body atoms. After substituting variables with constants, e.g. let $C_1, C_2, C_3$ be constants, $Pred_1(C_1, C_2) \wedge Pred_2(C_2, C_3) \Rightarrow Pred_3(C_1, C_3)$, which is called ground rule, and each atom with variables replaced by constants is called a ground predicate, whose value is a binary truth value. For example, $Pred_1(C_1, C_2) = \{0, 1\}$. If $Pred_1(C_1, C_2) \in F$, then $Pred_1(C_1, C_2) = 1$. Therefore, the goal of knowledge reasoning is to infer unknown facts $U = \{U_j\}$ from the known facts $F = \{f_i = 1\}_{i=1,2,\ldots}$

Inferring unknown facts from known facts is a generative problem, which requires building a joint probability distribution model and maximizing the generation probability to obtain the unknown facts. Hence, we must construct a suitable joint probability distribution model for the reasoning task. Considering the above conditions, the knowledge graph can be modeled as a MLN, which combines first-order predicate logic and probabilistic graphical models. Traditional methods employ first-order predicate logic for deductive reasoning in a black-and-white manner. However, as the example $S(Tom, basketball) \wedge F(Tom, John) \Rightarrow S(John, basketball)$ shows, it is not necessarily always true. MLN assign a weight $\omega$ to each rule, representing the probability of the event occurring, thus transforming the hard conditions of predicate logic into probabilistic conditions. The rule representation form in first-order predicate logic is converted to Conjunctive Normal Form (CNF) for computational convenience.

$$S(A, B) \wedge F(A, C) \Rightarrow S(C, B) \Leftrightarrow \neg S(A, B) \vee \neg F(A, C) \vee S(C, B)$$

Therefore, to construct a MLN from a knowledge graph, each ontology rule needs to be defined as a network in the MLN, each having a weight $\omega$. The probability calculation formula for MLN is

$$P(F, U|\omega) = \frac{1}{Z(\omega)} \prod_{r \in R} \exp\left(\omega_r N(F, U)\right), \tag{1}$$

where $F$ is the set of known facts, $U$ is the set of unknown facts, $R = \{r\}$ is the set of rules, $\omega_r$ is the weight of rule $r$, and $N(F, U)$ is the number of ground rules satisfying rule $r$. $Z(\omega)$ is the partition function, which is the sum of all possible ground rule cases for normalization

$$Z(\omega) = \sum_{F, U} \prod_{r \in R} exp(\omega_r N(F, U)). \tag{2}$$

All ground rules of each rule form a clique in MLN, $\exp\left(\omega_r N\left(F, U\right)\right)$ is the potential function of rule $r$, and each potential function expresses the situation of a clique. Generally, all ground rules of one rule form a clique, where each primary node, i.e., fact, is treated as a basic atom. Each state of MLN assigns different occurrence possibilities to all facts, representing a possible open world. Each set of possible worlds combines $\{F, U, R\}$ relations, jointly determining the truth values of all basic atoms. After establishing the joint probability distribution, we infer the unknown facts $U$ from the known facts $F$ by solving the posterior distribution $P(U|F, \omega)$, which can be viewed as an approximate inference problem.

Unlike rule-based reasoning methods that evaluate rules holistically, knowledge embedding methods mainly score facts, assigning higher scores to correct facts and lower scores to incorrect ones, obtaining embedding vectors for different entities, and enabling inference of unknown facts.

## 4 MODEL

This section introduces a knowledge reasoning method that combines MLNs with embedding learning. By utilizing MLN, which is trained with EM algorithm(Neal & Hinton, 1998), to establish a joint probability distribution model of known facts and unknown facts, we decompose $P(F|\omega)$ to obtain the following equation

$$logP(F|\omega) = log[\frac{P(F, U|\omega)}{Q(U)}] - log[\frac{P(U|F, \omega)}{Q(U)}], \tag{3}$$

where $P(F, U|\omega)$ is the joint probability distribution of known facts and unknown facts. In contrast, $P(U|F, \omega)$ is the posterior distribution, and $Q(U)$ is the approximate posterior distribution. Taking the expectation of both sides of equation(3) with respect to $Q(U)$, we can define $logP(F|\omega)$ as the sum of the evidence lower bound(ELBO) and the Kullback-Leibler(KL) divergence

$$logP(F|\omega) = ELBO + KL(q||p), \tag{4}$$

where $ELBO = \sum_U Q(U) log\left(\frac{P(F, U|\omega)}{Q(U)}\right), KL(q||p) = -\sum_U Q(U) log\left(\frac{P(U|F, \omega)}{Q(U)}\right).$

When the approximate posterior distribution $Q(U)$ is the same as the true posterior distribution, we obtain the optimal result, at which point $KL(q||p)$ is 0 and ELBO is maximized. Therefore, our optimization objective becomes maximizing the ELBO value

$$d_{ELBO}(Q, P) = \sum_U Q(U) log P(F, U|\omega) - \sum_U Q(U) log Q(U), \tag{5}$$

the approximate posterior distribution $Q(U)$ is the probability distribution of unknown facts based on known facts.

Specifically, in the t-th iteration, the first step is to fix the rule weight $\omega$ as $\omega_t$, which is a constant. We then update the probability set of each factor in all ground rules through the reasoning method proposed in this paper and obtain the current approximate posterior probability distribution $Q(U)$.

The second step substitutes the approximate posterior distribution into ELBO and updates $\omega$ by maximizing ELBO

$$\omega = argmax_\omega \sum (Q(U)logP(F,U|\omega) - Q(U)logP(U,F|\omega^t)), \tag{6}$$

where the second term is independent of $\omega$ and can be treated as a constant. Therefore, to reduce computation, we simplify the first step to fixing $\omega$ and computing the expectation of $logP(F,U|\omega)$ concerning $Q(U)$. The second step fixes the posterior distribution and updates $\omega$, obtaining $\omega^{t+1} = argmax_\omega \sum_U Q(U)logP(F,U|\omega)$.

## 4.1 SCORING MODULE

The most crucial part of the entire reasoning architecture is generating the approximate posterior probability. We design a scoring module to generate evaluation scores for facts. The generated evaluation scores can be the approximate posterior probability to compute the KL divergence from the actual posterior distribution. Additionally, they must satisfy the constraint that the loss for correct facts is minimized while the loss for incorrect facts is maximized. Therefore, we use vectors $e_h$ and $e_t$ to represent the head and tail entity features in a fact while representing the relation using three weight matrices.

Our scoring module consists of three parts. First, an embedding network initializes the vector features for each entity. Then, a scoring function $g(l, e_h, e_t)$ computes the evaluation score for each fact. Finally, the evaluation scores are processed to form the approximate posterior probability. For the scoring function, the model computes the following function to represent the possibility of the head and tail entities forming a valid fact under a given relation

$$g(l, e_h, e_t) = u_R^T f(e_h^T W_R e_t + V_R \begin{bmatrix} e_h \\ e_t \end{bmatrix} + b_R), \tag{7}$$

where $f$ is a non-linear activation function. $W_R$ is a $d*d*k$ dimensional tensor, and $e_h^T W_R e_t$ results from a bilinear tensor product, yielding a k-dimensional vector. $V_R$ is a $k*2d$ dimensional tensor, and $V_R \begin{bmatrix} e_h \\ e_t \end{bmatrix}$ is the result of a linear tensor product, also a k-dimensional vector. $u_R$ and $b_R$ are also k-dimensional, so the final result is a scalar. We design the each parts as follows:

We set up initial vectors for entities in the knowledge graph separately. We then build a neural network to update the vector features for all entities. The output of this part is the updated head and tail entity vectors $\{e_h, e_t\}$ with dimension $d$.

We initialize a bilinear neural network layer $W_R$ and two linear neural network layers $V_R, u_R$. Taking the head and tail entity vector features as input, we pass them through the scoring function $g(l, e_h, e_t)$ to output the result and compute the evaluation scores for all known facts, unknown facts, and negative sample facts.

We define the obtained evaluation scores as the approximate posterior probability for known and unknown facts. Specifically, we process the evaluation scores using the sigmoid function to bound them between 0 and 1, i.e., $p = sigmod(g(l, e_h, e_t))$, where $sigmod(.) = \frac{1}{1+\exp(.)}$.

## 4.2 E-STEP

In the expectation step, to solve for the unknown facts in the knowledge graph based on the known facts, we need to obtain the posterior distribution $P(U|F, \omega)$ of the unknown facts. This can be achieved by minimizing the KL divergence between the approximate and true posterior distributions. However, directly solving the joint probability distribution model established by MLN is highly complex. Therefore, this paper randomly samples batches of ground rules to form datasets, wherein the ground rules are approximately independent of each batch. By applying the mean-field theorem(Neal & Hinton, 1998), we define the approximate posterior distribution as the product of the probability distributions of the individual ground rules. The truth value of a ground rule is 1 when it holds and 0 when it does not, and the truth value of each ground rule is jointly determined by the truth values of its constituent facts. Therefore, we set the probability distribution of a ground rule

as the product of the probability distributions of its constituent facts. For example, for the ground rule: $R_1 = \neg S(Tom, basketball) \bigvee \neg F(Tom, John) \bigvee S(John, basketball)$.

The truth value of the ground rule R1 is determined by its three constituent facts. Thus, we define

$$Q(U) = \prod_{u_g \in U} q(u_g) = \prod_{u_g \in U} \prod_{u_k \in u_g} f_k(u_k), \tag{8}$$

where $u_k$ represents the value of fact $k$, which is either 0 or 1, where 1 indicates the fact holds and 0 indicates it does not. $u_g$ represents the set of all values of facts in an instance $g$ that belong to a rule and $U$ is the set of unknown facts. Each fact probability distribution $f_k(u_k)$ follows a Bernoulli distribution, where the truth value is 1 when the fact occurs and 0 when it does not, i.e., $f_k(u_k) = p_k^{u_k}(1 - p_k)^{(1-u_k)}$. The probability $p_k$ of the fact occurring is obtained from the scoring module.

The truth value of each ground rule is jointly determined by the truth values of its constituent facts. Therefore, the number of ground rules is represented as

$$N(F, U) = \sum_{u_g \in u_r} \prod_{u_k \in u_g} u_k, \tag{9}$$

where $u_r$ represents the set of facts belonging to rule $r$. Thus, equation (1) can be defined as

$$P(F, U|\omega) = \frac{1}{Z(\omega)} \prod_{r \in R} \exp\left(\omega_r \sum_{u_g \in u_r} \prod_{u_k \in u_g} u_k\right). \tag{10}$$

Substituting equations (8) and (10) into the optimization function (5), the term $Z(\omega)$ can be treated as a constant, leading to

$$\mathcal{L}_{ELBO} = \sum_{r \in R} \omega_r \sum_{u_g \in u_r} \prod_{u_k \in u_g} p_k - \sum_{r \in R} \sum_{u_g \in u_r} \sum_{u_k \in u_g} ((1 - p_k)\log(1 - p_k) + p_k log p_k). \tag{11}$$

This paper constructs the score $d_{fact}$ of the known fact set $F$ to add constraints.

$$d_{fact} = -\lambda \sum_F (\log(1 - p_k) + \log p_k). \tag{12}$$

We want the score $d_{fact}$ of the positive sample to be as small as possible. The final objective function is defined as

$$\mathcal{L} = \sum_{r \in R} \left(\omega_r \sum_{u_g \in u_r} \prod_{u_k \in u_g} p_k - \sum_{u_g \in u_r} \sum_{u_k \in u_g} ((1 - p_k)\log(1 - p_k) + p_k log p_k)\right) + d_{fact}. \tag{13}$$

### 4.3 M-STEP

In the M-step, we fix $Q(U)$ and then update the weights $\omega_r$ of the rule set $R$. At this point, the partition function in equation (2) from the E-step is no longer a constant. Therefore, in the M-step, we optimize the rule weights by minimizing the negative of the ELBO. However, when dealing with large-scale knowledge graphs, the number of facts also becomes enormous, making it difficult to optimize the ELBO directly. Consequently, we adopt the widely used pseudo-log-likelihood [39] as an alternative optimization objective, defined as

$$P(F, U|\omega) := \sum Q(U) \left(\sum_{u_k \in U} log P(u_k|\omega, MB_k)\right). \tag{14}$$

$MB_k$ represents the Markov Blanket of an individual fact $k$ in a ground rule. Therefore, following existing studies (Qu & Tang, 2019)(Zhang et al., 2020b), for each grounding formula $k$ connecting the base predicate with its Markov Blanket, we optimize the weights using the gradient descent formula

$$\nabla_{\omega_k} \sum f(u_k)(log P(u_k|\omega, MB_k)). \tag{15}$$

## 5 EXPERIMENT

### 5.1 EXPERIMENT SETTINGS

We evaluate the NPLL method on seven benchmark datasets through the knowledge base completion task and compare it with other state-of-the-art knowledge base completion methods. We show the code in supplementary material.

**Datasets.** We evaluate our proposed model on seven widely used benchmark datasets. Specifically, we use the YAGO3-10(Mahdisoltani et al., 2014) , YAGO37(Guo et al., 2018), Codex-L(Safavi & Koutra, 2020), UMLS(Bodenreider, 2004), Kinship (Hinton, 1990), FB15k-237 (Toutanova & Chen, 2015),WN18RR(Dettmers et al., 2018). YAGO3-10 is a subset of YAGO3 (an extension of YAGO) that contains entities associated with at least ten different relations. YAGO37 is also a variant of YAGO dataset. Codex-L is a set of knowledge graph Completion Datasets Extracted from Wikidata and Wikipedia. FB15k-237 and WN18RR are more challenging versions of the FB15K and WN18 datasets. The Unified Medical Language System (UMLS) is a comprehensive resource that integrates and disseminates essential terminology, classification standards, and coding systems. The Kinship dataset is a relational database consisting of 24 unique names in two families. Appendix A shows details of datasets.

**Evaluation metrics.** Following existing studies(Bordes et al., 2013), we use the filtered setting during evaluation. Mean Reciprocal Rank (MRR), Hit@10, Hit@3, and Hit@1 are treated as the evaluation metrics.

**Competitor methods**: We compare knowledge graph embedding methods, rule-based methods, and methods combining the two. For knowledge graph embedding methods, we select some of the most classic distance translation and semantic matching algorithms, including TransE(Bordes et al., 2013), DistMult(Yang et al., 2014), ComplEx (Trouillon et al., 2016), ConvE(Dettmers et al., 2018), RotatE(Sun et al., 2019). For rule-based reasoning algorithms that integrate rules, we compare with NeuralLP(Yang et al., 2017), DRUM(Sadeghian et al., 2019), pLogicNet(Qu & Tang, 2019), ExpressGNN(Zhang et al., 2020b), DiffLogic(Shengyuan et al., 2024), NCRL(Cheng et al., 2023). The comparative experiments are conducted under the same experimental conditions, selecting the best training hyperparameters provided by the open-source codes of each algorithm.

**Experimental setting**: For the selection of logical rules across the seven benchmark datasets, we first generated candidate rules using the Neural LP (Yang et al., 2017) method, a state-of-the-art rule mining approach. We then preprocessed the candidate rules by removing self-reflective logical rules and eliminating duplicates. Next, we applied a confidence score threshold, selecting all rules with a confidence score greater than a predefined parameter $\alpha$ for the same target predicate. Using the successive approximation method, we selected the optimal prior rules for each dataset, adjusting the number of approximation iterations based on the volume of candidate rules. For instance, as shown in Figure 2, we demonstrate the process of obtaining the optimal rules for the YAGO3-10 dataset through three rounds of approximation experiments, ultimately choosing a set of rules with confidence scores exceeding 0.341.

Finally, we determined the most suitable logical rule set for each dataset through extensive experiments. Based on the experimental results, we identified the optimal rule sets for each dataset as follows: for the YAGO3-10 dataset, we selected rules with a confidence score greater than 0.341; for the UMLS datasets, rules with a confidence score greater than 0; for the Codex-L dataset, rules with a confidence score greater than 0.61; for the Fb15k-237 dataset, rules with a confidence score greater than 0.87;for the YAGO37, Kinship and WN18RR datasets, rules with a confidence score greater than 1.0. This systematic process of rule selection and empirical evaluation allowed us to identify the most suitable logical rules for each knowledge graph, ensuring that our proposed method leverages high-quality symbolic knowledge to enhance its reasoning capabilities.

**General setting**: All experiments are conducted on the same server with two GPUs (Nvidia RTX 3090, 24G), using Cuda version 11.8, Ubuntu 22.04.6 system, and Intel(R) Xeon(R) CPU E5-2620 v3 @ 2.40GHz CPU.

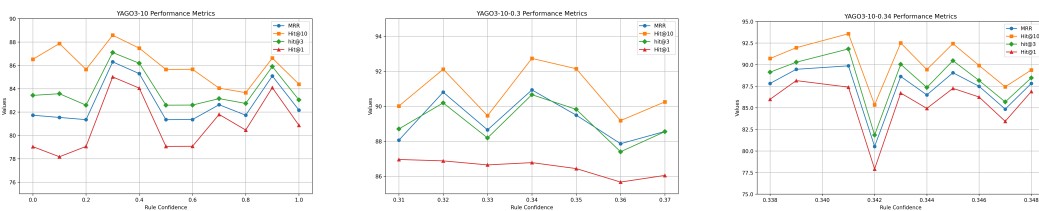

Figure 2: Logic rule generation by successive approximation method

## 5.2 RESULTS

**Large scale KG completion performance analysis.** The experimental results are presented in Tables 1. We have organized our findings based on the scale of the knowledge graphs under investigation.Appendix A shows the experimental outcomes for three large-scale datasets: YAGO3-10, YAGO37, and Codex-L. The first two datasets encompass millions of training facts, while Codex-L comprises over 500,000 training instances.Our analysis reveals that both variants of the NPLL method demonstrated robust performance across all datasets. Notably, NPLL-basic significantly outperformed other baseline methods on large-scale datasets. The Hit@1 and Hit@3 scores for NPLL closely approximate its Hit@10 score, indicating a substantial enhancement in the quality of inferred results.

Table 1: Results of large KG completion. We select the metrics provided in the papers for the DiffLogic and NCRL algorithms from the rule-learning methods, as we could not find suitable open-source codes for them. [NA] indicates that the model cannot finish inference in our machines.The red numbers indicate the best performance achieved on a particular metric. Hit@K is in %.

| Methods | Models | YAGO3-10 | | | | YAGO37 | | | | Codex-L | | | |
|---|---|---|---|---|---|---|---|---|---|---|---|---|---|
| | | MRR | Hit@10 | Hit@3 | Hit@1 | MRR | Hit@10 | Hit@3 | Hit@1 | MRR | Hit@10 | Hit@3 | Hit@1 |
| KGE | TransE | 0.4216 | 65.19 | 52.16 | 28.39 | 0.4090 | 63.94 | 51.94 | 26.80 | 0.2097 | 39.78 | 29.09 | 9.24 |
| | DistMult | 0.3330 | 52.80 | 32.21 | 24.15 | 0.4062 | 57.61 | 45.19 | 31.91 | 0.2578 | 36.18 | 28.32 | 20.17 |
| | ComplEx | 0.3465 | 54.75 | 24.15 | 16.30 | 0.4247 | 58.11 | 46.91 | 34.37 | 0.2866 | 39.82 | 31.44 | 22.64 |
| | RotatE | 0.4913 | 67.10 | 54.52 | 39.81 | 0.4361 | 61.29 | 48.16 | 34.62 | 0.2870 | 39.49 | 31.44 | 22.88 |
| Rule-Learning | Neural LP | NA | NA | NA | NA | NA | NA | NA | NA | 0.1244 | 16.12 | 13.13 | 10.16 |
| | pLogicNet | 0.2984 | 27.36 | 33.02 | 25.17 | 0.1095 | 14.73 | 11.83 | 8.62 | 0.1093 | 20.26 | 12.04 | 6.25 |
| | ExpressGNN | NA | NA | NA | NA | NA | NA | NA | NA | 0.0261 | 5.61 | 1.88 | 0.67 |
| | NCRL | 0.3800 | 53.60 | - | 27.40 | - | - | - | - | - | - | - | - |
| | DiffLogic | 0.5130 | 67.40 | - | - | - | - | - | - | 0.3370 | 46.00 | - | - |
| us | NPLL-basic | 0.8986 | 93.58 | 91.82 | 87.39 | 0.7023 | 74.81 | 71.43 | 67.72 | 0.7063 | 82.09 | 74.90 | 64.39 |
| | NPLL-GNN | 0.6201 | 77.72 | 66.99 | 53.75 | 0.4379 | 55.64 | 47.25 | 37.41 | 0.4837 | 63.46 | 51.48 | 40.83 |

**KG completion performance analysis.** The experimental results are shown in Table 2. The NPLL-basic and NPLL-GNN methods achieve good performance across all four datasets. On the FB15k-237 and UMLS datasets, the NPLL-basic method significantly outperforms other methods, achieving the best results on all four metrics. On the WN18RR and Kinship dataset, NPLL-basic and NPLL-GNN comprehensively outperform the data-driven embedding methods, while NPLL-basic achieve the best results on the MRR, Hit@3, and Hit@1 metrics. This indicates that the reasoning effectiveness and expressiveness of NPLL have been enhanced.

**Ablation study.** For our method, we consider two variants: NPLL-GNN, which utilizes a tunable graph neural network(Zhang et al., 2020b) in the scoring module for training, and NPLL-basic, which employs only a single-layer embedding network in the scoring module for training. We examine how different representations of entities and relations affect the performance of our NPLL model. By systematically varying the embedding strategies, we aim to understand their contributions to the model's inferential capabilities. Our comprehensive ablation analysis spans all datasets, allowing us to draw robust conclusions about the relationship between embedding choices and predictive accuracy. The comparative outcomes of two distinct embedding methodologies applied within the NPLL framework are presented in Table 1 and Table 2, providing insights into their relative effectiveness across various knowledge graph scenarios. Compared to other baseline methods, both NPLL-basic and NPLL-GNN perform excellently across all datasets, with NPLL-basic generally achieving better results. Only on the smaller UMLS datasets does NPLL-GNN score similarly or

Table 2: Results of KG completion. We select the metrics provided in the papers for the DiffLogic and NCRL algorithms from the rule-learning methods, as we could not find suitable open-source codes for them. (The red numbers indicate the best performance achieved on a particular metric.) Hit@K is in %.

| Methods | Models | FB15k-237 | | | | WN18RR | | | |
|---------|--------|-----------|--------|-------|-------|--------|--------|-------|-------|
| | | MRR | Hit@10 | Hit@3 | Hit@1 | MRR | Hit@10 | Hit@3 | Hit@1 |
| KGE | TransE | 0.33 | 52.71 | 29.28 | 18.93 | 0.2231 | 52.12 | 40.10 | 1.31 |
| | DistMult | 0.2878 | 45.67 | 31.43 | 20.31 | 0.4275 | 50.71 | 44.01 | 38.21 |
| | ComplEx | 0.3016 | 48.08 | 33.10 | 21.28 | 0.4412 | 51.03 | 46.11 | 41.01 |
| | ConvE | 0.3251 | 50.11 | 35.68 | 23.80 | 0.4295 | 52.13 | 44.34 | 39.87 |
| | RotatE | 0.3213 | 53.10 | 34.52 | 22.81 | 0.4714 | 55.71 | 47.29 | 42.87 |
| Rule-Learning | Neural LP | 0.1983 | 29.84 | 21.73 | 14.48 | 0.3800 | 40.79 | 38.81 | 36.80 |
| | DRUM | 0.2430 | 36.39 | 21.91 | 17.43 | 0.3861 | 41.02 | 38.93 | 36.91 |
| | pLogicNet | 0.3300 | 52.79 | 36.87 | 23.12 | 0.2300 | 53.09 | 41.48 | 1.5 |
| | ExpressGNN | 0.4894 | 60.80 | 48.10 | 38.91 | - | - | - | - |
| | NCRL | 0.3000 | 47.30 | - | 20.90 | 0.6700 | 85.00 | - | 56.30 |
| | DiffLogic | - | - | - | - | 0.5001 | 58.70 | - | - |
| us | NPLL-basic | 0.6223 | 68.57 | 64.52 | 58.63 | 0.7668 | 78.14 | 77.38 | 75.83 |
| | NPLL-GNN | 0.5442 | 61.93 | 57.06 | 50.25 | 0.5282 | 61.52 | 55.50 | 48.17 |

| Methods | Models | Kinship | | | | UMLS | | | |
|---------|--------|---------|--------|-------|-------|------|--------|-------|-------|
| | | MRR | Hit@10 | Hit@3 | Hit@1 | MRR | Hit@10 | Hit@3 | Hit@1 |
| KGE | TransE | 0.3509 | 80.36 | 50.14 | 1.10 | 0.7806 | 99.13 | 96.05 | 59.56 |
| | DistMult | 0.3925 | 77.86 | 42.68 | 23.73 | 0.4770 | 78.83 | 53.87 | 33.57 |
| | ComplEx | 0.7201 | 95.91 | 80.86 | 59.73 | 0.8950 | 98.34 | 95.58 | 82.70 |
| | RotatE | 0.4890 | 86.95 | 56.32 | 32.41 | 0.5884 | 83.41 | 68.33 | 44.23 |
| Rule-Learning | Neural LP | 0.5637 | 88.00 | 63.94 | 41.49 | 0.7312 | 91.29 | 84.70 | 59.37 |
| | DRUM | 0.3312 | 70.15 | 48.23 | 25.67 | 0.5634 | 85.64 | 65.58 | 35.79 |
| | NCRL | 0.6400 | 92.90 | - | 49.00 | 0.7800 | 95.10 | - | 65.90 |
| us | NPLL-basic | 0.8663 | 92.68 | 87.91 | 83.55 | 0.9763 | 99.21 | 98.26 | 96.76 |
| | NPLL-GNN | 0.7705 | 87.55 | 79.09 | 71.77 | 0.9754 | 99.05 | 98.66 | 96.45 |

Table 3: A comparison of the model parameter counts for NPLL-basic, NPLL-GNN, and ExpressGNN methods on the FB15k-237 dataset

| | Models | FB15k-237 |
|---|--------|-----------|
| | ExpressGNN | 251,337k |
| Total params count(k) | NPLL-basic | 64,967k |
| | NPLL-GNN | 64,953k |

slightly better on Hit@10, Hit@3, Hit@1 and MRR. This indicates that NPLL-GNN approaches NPLL-basic in expressiveness on the UMLS dataset. We hypothesize that due to the characteristics of the GNN network, it can better transmit information and extract features on complex networks. The UMLS dataset has comprehensive logic rules, allowing the construction of information-rich Markov logic networks, thereby enhancing the expressiveness of NPLL-GNN on such data.

**Parameter counts.** The terms of model parameter counts, we compare NPLL with the ExpressGNN method, which has relatively high overall performance among the baseline methods on the FB15k-237 dataset. As shown in Table 3, the parameter count of our method is approximately one-fourth of ExpressGNN.

**Analysis of data efficiency.** We investigate the data efficiency of NPLL-basic and NPLL-GNN, and compare them with baseline methods. We divide the FB15k-237 knowledge base into fact/train/valid/test files(Yang et al., 2017), and vary the size of the train set from 0% to 20%, while providing the complete fact set to the models. The results can be seen in Table 4. In Figures 3,

Table 4: Results on the FB15k-237 dataset with various data sizes. Hit@K is in %

| Models | FB-0 | | | | FB-0.05 | | | |
|---|---|---|---|---|---|---|---|---|
| | MRR | Hit@10 | Hit@3 | Hit@1 | MRR | Hit@10 | Hit@3 | Hit@1 |
| TransE | 0.2412 | 42.71 | 26.39 | 16.10 | 0.2523 | 43.09 | 26.87 | 16.43 |
| Neural LP | 0.0128 | 1.75 | 0.73 | 0.41 | 0.1531 | 24.51 | 16.72 | 10.43 |
| DistMult | 0.2297 | 38.87 | 25.02 | 15.10 | 0.2317 | 39.28 | 25.13 | 15.25 |
| CompIEx | 0.2363 | 40.29 | 25.72 | 15.47 | 0.2395 | 40.70 | 25.98 | 15.75 |
| ExpressGNN | 0.4276 | 53.88 | 45.74 | 36.65 | 0.4187 | 54.24 | 44.89 | 35.50 |
| NPLL-basic | 0.5356 | 62.55 | 56.87 | 51.03 | 0.5384 | 63.09 | 57.84 | 51.38 |
| NPLL-GNN | 0.4989 | 58.78 | 52.95 | 44.88 | 0.4911 | 58.15 | 52.23 | 44.07 |

| Models | FB-0.1 | | | | FB-0.2 | | | |
|---|---|---|---|---|---|---|---|---|
| | MRR | MRR | MRR | MRR | MRR | Hit@10 | Hit@3 | Hit@1 |
| TransE | 0.2531 | 43.41 | 26.92 | 16.68 | 0.2533 | 43.92 | 27.13 | 16.81 |
| Neural LP | 0.1624 | 25.88 | 17.81 | 11.16 | 0.1699 | 26.79 | 18.53 | 11.86 |
| DistMult | 0.2333 | 39.47 | 25.36 | 15.37 | 0.2371 | 40.07 | 25.80 | 15.64 |
| CompIEx | 0.2409 | 40.74 | 26.24 | 15.89 | 0.2451 | 41.63 | 26.71 | 16.16 |
| ExpressGNN | 0.4226 | 55.30 | 45.49 | 35.91 | 0.4273 | 55.59 | 45.81 | 36.34 |
| NPLL-basic | 0.5466 | 63.40 | 57.20 | 51.93 | 0.5594 | 63.62 | 57.57 | 52.11 |
| NPLL-GNN | 0.5241 | 59.66 | 54.85 | 48.33 | 0.5307 | 60.55 | 55.69 | 48.91 |

the NPLL methods are shown as solid lines, while other methods are dashed lines. We can clearly see that NPLL performs significantly better than the baselines with smaller training data. Even with more training data for supervision, NPLL still exhibits excellent performance across all metrics. This clearly demonstrates that NPLL can more accurately predict the correct answers and has outstanding data utilization ability.

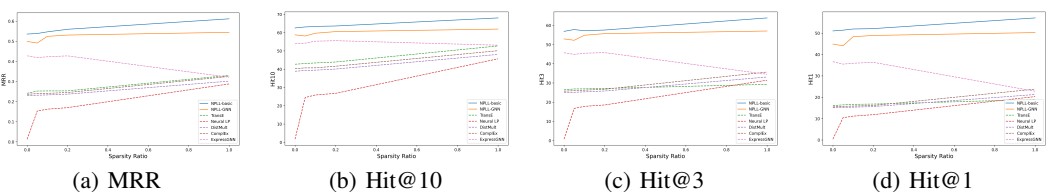

(a) MRR          (b) Hit@10          (c) Hit@3          (d) Hit@1

Figure 3: Performance of KG completion vs sparsity ratio

## 6 CONCLUTION

In this paper, we study knowledge graph reasoning and propose a method called Neural Probabilistic Logic Learning (NPLL), which effectively integrates logical rules with data embeddings. NPLL utilizes neural networks to extract node features from the knowledge graph and then supports the reasoning of Markov Logic Networks through a scoring module, effectively enhancing the model's expressiveness and reasoning capabilities. NPLL is a general framework that allows tuning the encoding network to boost model performance.

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

APPENDIX

# A DATASET DETAILS

To comprehensively evaluate the performance of our proposed method, we conducted extensive comparative experiments across seven widely-adopted benchmark datasets: YAGO3-10, YAGO37, Codex-L UMLS, Kinships, FB15k-237, and WN18RR. Additionally, to investigate the impact of dataset size on reasoning performance, we performed a splitting operation on the FB15k-237 dataset, creating four subsets: FB-0, FB-0.05, FB-0.1, and FB-0.2, where the Train file was divided into varying proportions. The specific details and statistics of these datasets are provided in Table 5.

This diverse set of benchmark datasets allows for a comprehensive evaluation of our method's reasoning capabilities across varying dataset sizes, knowledge graph complexities. The YAGO3-10, YAGO37 and Codex-L represent large scale knowledge graphs,The UMLS and Kinships datasets represent domain-specific knowledge graphs, while FB15k-237 and WN18RR are more general-purpose knowledge bases. By including both small-scale and large-scale datasets, we can thoroughly assess the robustness, scalability, and generalization abilities of our proposed approach under a wide range of conditions encountered in real-world knowledge graph reasoning tasks.

Table 5: Knowledge base completion datasets statistics

| Dataset | #Fact | #Train | #Test | #Valid | #Relation | #Entity | #Rules |
|---------|-------|--------|-------|--------|-----------|---------|--------|
| YAGO3-10 | 809280 | 269760 | 4982 | 4978 | 37 | 123182 | 348 |
| YAGO37 | 741849 | 247283 | 50000 | 50000 | 37 | 123189 | 115 |
| Codex-L | 413394 | 137799 | 30622 | 30622 | 69 | 77951 | 300 |
| Fb15k-237 | 204087 | 68028 | 20466 | 17536 | 237 | 14541 | 516 |
| Fb-0 | 204087 | 1 | 20466 | 17536 | 237 | 14541 | 516 |
| Fb-0.05 | 204087 | 3401 | 20466 | 17536 | 237 | 14541 | 516 |
| Fb-0.1 | 204087 | 6802 | 20466 | 17536 | 237 | 14541 | 516 |
| Fb-0.2 | 204087 | 13605 | 20466 | 17536 | 237 | 14541 | 516 |
| WN18RR | 65127 | 21708 | 3134 | 3034 | 11 | 40943 | 33 |
| Kinship | 6375 | 2112 | 1100 | 1099 | 25 | 104 | 71 |
| UMLS | 4006 | 1321 | 633 | 569 | 46 | 135 | 1055 |

# B TRAINING TIME DETAILS

Table 6 details more aspects of the training time.

Table 6: Total train time of KG completion

| Models | yago37 | YAGO3-10 | Codex-L | FB15k-237 | Kinship | WN18RR | UMLS |
|--------|--------|----------|---------|-----------|---------|--------|------|
| NPLL-basic | 5214s | 2301s | 10282s | 2690s | 383s | 198s | 1816s |

