# OpenReview forum: "Neural Probabilistic Logic Learning for Knowledge Graph Reasoning"
_ICLR.cc/2025/Conference — ICLR 2025 Conference Withdrawn Submission_

### Official Review · Reviewer_1oQB · 2024-10-26

**Soundness:** 1
**Presentation:** 1
**Contribution:** 1
**Rating:** 1
**Confidence:** 3

**Summary:**

The paper studies the popular problem of knowledge graph completion. The paper contrasts rule based methods, on the one hand, with embedding based methods, on the other hand. Rules are said to be more accurate but less efficient, while embeddings are said to be less accurate and more efficient. The paper therefore proposes a strategy for combining the advantages of both, by using a variational approximation of Markov logic networks.

**Strengths:**

Studying new methods in which embedding and rule based methods can be combined is clearly of interest.

The experimental results are very good, spectacular even.

**Weaknesses:**

The core idea of this paper is identical to that of the pLogicNet paper, which also proposes a variational approximation of Markov logic networks based on embeddings. Furthermore, ExpressGNN builds on pLogicNet by using GNNs instead of embeddings, and this paper similarly analyses a variant based on GNNs. It is not clear what is novel about the proposed model compared to these two earlier models. Worryingly, while pLogicNet and ExpressGNN are cited in the paper, no mention at all is made of the close correspondence. All that is said about pLogicNet, for instance, is that it is a "probabilistic logic reasoning network ... demonstrating exemplary performance".  If there is a conceptual difference with pLogicNet which I missed, the paper should have discussed this explicitly.

The experimental results are substantially better than those of existing models, including those of pLogicNet. Given the close similarity with pLogicNet, this makes the validity of the results questionable. At a minimum, the paper should have analysed the differences with pLogicNet. For instance, to what extent can the performance differences be explained by the fact that both methods start from a different set of rules? If this does not explain the difference, then what is responsible for this huge performance gap?

The GNN variant of the model is introduced while discussing the experimental results, but is never properly explained. Table 3 shows that it has only a quarter of the parameters of ExpressGNN, but without further details, this seems to be a matter of different hyper parameter tuning, rather than any genuine difference.

The paper is poorly written. For instance, even the motivation in the introduction doesn't really make sense. Knowledge graphs are said to capture "rich semantics" and offer "more expressive" representations than traditional methods, which doesn't make sense to me. The technical details, for instance in Section 4.1, are very hard to follow.

**Questions:**

Why did you not discuss the close similarity with pLogicNet and ExpressGNN in the paper?

Why did you not analyse where the performance improvements compared to pLogicNet and ExpressGNN are coming from, given the very close similarity with these models?

---

### Official Review · Reviewer_VfE1 · 2024-10-30

**Soundness:** 2
**Presentation:** 1
**Contribution:** 2
**Rating:** 3
**Confidence:** 4

**Summary:**

This paper proposes a rule-based KG reasoning method. The main idea is based on markov logic network. Compared with baseline method, the proposed method is more effective and performs the best over several benchmarks. In particular, the proposed NPLL is effective in data-scarse scenarios.

**Strengths:**

1. The proposed method outperforms the baseline methods by a large margin.
2. The proposed method also performs well in the data-scarce cases.
3. The proposed method is parameter efficient.
4. Code is provided.

**Weaknesses:**

1. This paper is ill-written. The motivation is extremely unclear. Over the entire paper, it is hard for me to capture what problem this paper want to address and how the proposed method is motivated.
2. The advantage of NPLL over other methods is not well discussed. The authors frequently claim that NPLL is more effective than the others. However, in what aspects and why? I can't understand.
3. As for the methodology, there lack of an overall picture of the whole framework. Just talk about what they do with many details for about 3.5 pages. The visualization of the framework in Figure 1 is also not well illustrated.
4. For the contributions, I don't know why the second and third properties are important. In other words, how these properties or designs benefit the KG reasoning problem is not clear.
5. Section 2 mentions interpretability, but experimental results do not show this point.
6. There are many typos and inappropriate expressions. For example,
- Section 3 uses mixed expressions of italic and normal fonts, e.g., E and $E$, L and $L$, fi and $f_i$.
- In line 142, what is $y_i'$?
- exp in Equations 1 and 2 with different fonts.
7. The font sizes in figure 2 and 3 are too small.

**Questions:**

Please check the questions in weakness points.

---

### Official Review · Reviewer_WXKG · 2024-11-01

**Soundness:** 2
**Presentation:** 1
**Contribution:** 2
**Rating:** 3
**Confidence:** 3

**Summary:**

1. The paper introduces Neural Probabilistic Logic Learning (NPLL), a rule-based method for knowledge graph reasoning. NPLL represents knowledge using a Markov Logic Network (MLN), enhancing the expressiveness of embedding networks. Through variational inference, NPLL accurately infers unknown facts and introduces a scoring module to improve reasoning accuracy in knowledge graphs.

2. The authors conduct extensive experiments across various benchmark datasets, including YAGO3-10, YAGO37, Codex-L, WN18RR and FB15k-237. The results demonstrate that NPLL achieves superior reasoning performance, surpassing other methods on large-scale and domain-specific datasets. This validation highlights NPLL’s effectiveness in complex knowledge graph reasoning tasks.

**Strengths:**

1. The methodological improvement introduced in this paper that leverages an embedding-based scoring module is straightforward. However, this simplicity contributes to the model’s robustness and ease of implementation, and the results achieved are notably impressive.

2. The experimental evaluation is thorough, covering a wide range of benchmark knowledge graphs. This comprehensive testing approach not only underscores the model’s versatility but also consistently demonstrates superior performance across diverse datasets, reinforcing the robustness and broad applicability of the proposed method.

**Weaknesses:**

## 1 Novelty Issue
1. The proposed methodology closely resembles the approach used in ExpressGNN [1], with the primary difference being the addition of a scoring module on factual triples $(e_h, l, e_t)$. It remains unclear what further distinctions, if any, exist between this model and ExpressGNN, raising concerns regarding the novelty of this contribution.

## 2 Insufficient and Unjustified Experimentation
1. Two model variants, NPLL-basic and NPLL-GNN, are proposed, with reasoning results provided for each (Table 2). However, no explanation is given for the significantly lower accuracy of NPLL-GNN compared to NPLL-basic, leaving questions about model performance unaddressed.

2. The evaluation of data efficiency (Table 4) is limited to the FB15k-237 dataset, replicating results that have already been demonstrated in ExpressGNN. Since this work (NPLL) and ExpressGNN [1] follow the same framework (MLN), results on additional datasets would help clarify the model’s data efficiency. Additionally, the impact of varying data sizes on NPLL-basic performance is minimal (Table 4), but this observation is neither analyzed nor discussed.

3. In the Related Work section, the authors claim that compared to embedding-based methods, NPLL enhances both interpretability and reasoning quality. However, no experimental evidence is provided to substantiate this claim.

## 3 Representation
1. Improper use of mathematical symbols: The mathematical formulations are often imprecise, with symbols inconsistently defined or unclear. For example, the definitions of “fact” in the preliminary section are ambiguous, and the MLN setup and representation of unknown facts are incomplete. Including concrete examples would improve clarity and reader comprehension. In the Model section, certain notations (e.g., $u_g$, $u_k$ in Equation 8) are confusing and inadequately defined, making this section challenging to follow.

2. The text and curves in the all figures, especially Figure 2 and 3,  are difficult to read due to their small size, limiting accessibility to critical information.

3. Significant details are missing in the methods section, such as specifics on model training and the E-step and M-step processes. This lack of detail, particularly compared to ExpressGNN, further underscores the concerns about novelty in this work.

[1] Zhang, Y.; Chen, X.; Yang, Y.; Ramamurthy, A.; Li, B.; Qi, Y.; Song, L. Efficient Probabilistic Logic Reasoning with Graph Neural Networks. arXiv February 4, 2020. https://doi.org/10.48550/arXiv.2001.11850.

**Questions:**

1. Novelty: Beyond the addition of the scoring module, what are the other key differences between this approach and ExpressGNN? It would be helpful if the authors could elaborate on any unique elements or improvements this model brings, especially regarding interpretability, efficiency, or theoretical grounding.

2. Impact of different data sizes on model performance: The experiments with varying data sizes in the FB15k-237 dataset reveal minimal impact on model performance, but the reasoning behind this result is not addressed. Could the authors provide an analysis of why this might be the case? It would also be valuable if additional insights could be given on the robustness of the model in data-scarce environments or if alternative metrics could show nuanced performance variations.

3. Training time comparisons: Since NPLL utilizes rules derived from a pre-trained Neural-LP model on specific datasets, I suggest that the reported training time should also include the time required to train the Neural-LP model initially (Table 6).

---

### Official Review · Reviewer_4F7f · 2024-11-01

**Soundness:** 3
**Presentation:** 2
**Contribution:** 2
**Rating:** 5
**Confidence:** 4

**Summary:**

This paper explores the combination of embedding-based methods with rule-based reasoning approaches to address their individual shortcomings when used separately. The authors introduce Neural Probabilistic Logic Learning (NPLL), a framework that combines the strengths of embeddings and rule-based reasoning, which purportedly achieves efficient, large-scale knowledge graph reasoning. NPLL seemingly demonstrates high reasoning performance, even in data-scarce conditions, balancing model size with reasoning capability to enable practical applications.

**Strengths:**

This paper is generally well-written, though the clarity of some parts could be improved.

The experiments purportedly show huge gains by the paper's approach on standard benchmarks.

**Weaknesses:**

This paper does not clearly position its technical contributions with respect to prior research. Its approach resembles that of pLogicNet (Qu and Tang, 2019) and ExpressGNN (Zhang et al., 2020), both of which the paper cites. Both pLogicNet and ExpressGNN employ the same variational-EM approach as this paper, as well as the computational simplifications of mean-field approximation and pseudo-log-likelihood optimization. In light of these existing works, the primary contribution of this paper appears to be the introduction of the triplet scoring function in Equation 7, which is essentially a simple feed-forward network. Relative to prior work, this contribution seems incremental at best. Furthermore, in the context of existing embedding-based approaches, the triplet scoring function (Equation 7) may be redundant, as current embedding-based methods can produce scores using the same inputs (embeddings for entities and relations).

Although the empirical results show significant improvements over baseline scores, the paper’s exposition does not clearly explain how these improvements are achieved through the authors’ approach.

**Questions:**

Line 101: The paper claims that it "is significantly more effective for knowledge graph reasoning" than the prior works mentioned in lines 84-101 without clearly describing how it is technically superior. Could the authors please elaborate on their technical contributions relative to each of the related works?

Line 187, Section 4:
Could the authors please compare and contrast their NPLL model with the closely related models pLogicNet and ExpressGNN? Which specific components introduced in the paper account for the stellar empirical results reported?

Line 324, Section 5:
Which empirical results in this section support the authors' answer to the above question regarding the most effective components of their approach?

---

### Note · Authors · 2024-11-17

I have read and agree with the venue's withdrawal policy on behalf of myself and my co-authors.